# Uterine Disease in Dairy Cows: A Comprehensive Review Highlighting New Research Areas

**DOI:** 10.3390/vetsci11020066

**Published:** 2024-02-02

**Authors:** Zsóka Várhidi, György Csikó, Árpád Csaba Bajcsy, Viktor Jurkovich

**Affiliations:** 1Department of Animal Hygiene, Herd Health and Mobile Clinic, University of Veterinary Medicine, 1078 Budapest, Hungary; 2Department of Pharmacology and Toxicology, University of Veterinary Medicine, 1078 Budapest, Hungary; csiko.gyorgy@univet.hu; 3Clinic for Cattle, University of Veterinary Medicine Hannover, Foundation, 30173 Hannover, Germany; csaba.bajcsy@tiho-hannover.de; 4Centre for Animal Welfare, University of Veterinary Medicine, 1078 Budapest, Hungary

**Keywords:** uterine disease microbiome, probiotics, dairy cow

## Abstract

**Simple Summary:**

This literature review summarizes recent knowledge on uterine diseases in dairy cows. Metritis and endometritis are major problems in intensive dairy farming at present and cause significant long-term economic losses. Therefore, the efficient treatment and prevention of uterine diseases in the peri- and postpuerperal periods are essential. The use of antibiotics in these disease conditions is limited, either due to their limited effectiveness or due to recent legislation intended to reduce the use of antimicrobials. Therefore, finding efficient alternatives to antibiotics is necessary. This literature review provides an overview of the current and future possibilities for the non-antibiotic-based control of metritis and endometritis in dairy cows.

**Abstract:**

Uterine disease is an intensely studied part of dairy cattle health management as it heavily affects many commercial dairy farms and has serious economic consequences. Forms of the disease, pathophysiology, pathogens involved and the effects of uterine disease on the health and performance of cows have already been well described by various authors. Lately, researchers’ attention has shifted towards the healthy microbiome of the uterus and the vagina to put emphasis on prevention rather than treatment. This aligns with the growing demand to reduce the use of antibiotics or—whenever possible—replace them with alternative treatment options in farm animal medicine. This review provides a comprehensive summary of the last 20 years of uterine disease research and highlights promising new areas for future studies.

## 1. Introduction

Uterine disease is still undoubtedly a major concern on most high-producing dairy cattle farms, as it affects about half of all dairy cows and has serious consequences regarding reproductive performance and milk production [1]. Problems with reproductive performance are the main reasons for culling on large-scale dairy cattle farms [2,3,4]. The average proportion of reproductive culling is around 30% of all premature disposals [5]. The economic costs of uterine disease consist of a lower milk yield, decreased pregnancy rate, higher chances of premature culling and increased replacement costs [6,7]. Although the disease itself, the economic consequences and the treatment options have been continuously studied in the past couple of years, there has not been a comprehensive summary of recent findings since the early 2000s [8]. The aim of this review is to provide researchers and veterinarians with an up-to-date review of research published during the last 20 years and show the most promising areas of related research in the upcoming years, paying special attention to intravaginal probiotics.

## 2. Uterine Disease of Dairy Cows

### 2.1. Forms of Uterine Disease of Dairy Cows

#### 2.1.1. Metritis: First 21 Days after Calving

Metritis is defined as the inflammation of the uterine wall, including the endometrium, the muscular layers and the serosa, which occurs until the first 21 days after calving—primarily within 7–10 days—and can affect up to 40% of dairy cows. Clinical signs include a watery, red-brown uterine discharge, usually with a fetid odour. The uterus of a metritic cow is enlarged and flaccid and does not have the longitudinal folds that typically characterize involution. Metritis is classified into three grades based on the accompanying systemic symptoms. In Grade 1 metritis, there is no sign of systemic signs of illness or fever. In Grade 2, or acute puerperal metritis, clinical signs include fever (>39.5 °C), reduced appetite, decreased milk production and prolonged resting periods (lethargy). In Grade 3, or toxic metritis, the cow is recumbent and shows toxaemic signs [9,10,11].

#### 2.1.2. Endometritis: From 21 Days after Calving

Endometritis is defined as inflammation of the endometrium only that does not extend beyond the stratum spongiosum. Endometritis occurs from 21 days after calving onwards and is classified as clinical or subclinical [12]. The disease can affect about 20% of dairy cows [13,14,15].

Clinical endometritis is characterized by visible mucopurulent (50% mucus; 50% pus) or purulent discharge at the vulva or in the vagina. The severity of clinical endometritis is usually graded by evaluating the vaginal discharge. In Grade 0, or normal (without endometritis), the discharge is clear or translucent. In Grade 1, the mucoid discharge contains flecks of white or off-white pus. In Grade 2, the discharge contains less than 50% white or off-white mucopurulent material. In Grade 3, it is purulent, usually with a white or yellow colour, but sometimes it can contain blood too [9,16]. In subclinical endometritis, the infection and inflammation of the endometrium do not result in discharge.

#### 2.1.3. Pyometra: Between 42 and 60 Days after Calving

Pyometra is sometimes also considered a type of endometritis; however, usually, it is discussed separately [8,10]. It is characterized by the accumulation of pus in the distended uterus with a closed cervix and a persistent corpus luteum in an ovary. Some pus my leak into the vagina too. Bacterial infection of the oviducts and salpingitis can occur and influence fertility as well. Pyometra occurs between 42 and 60 days after calving, with a relatively low prevalence rate of less than 2% [8,9,17,18,19].

### 2.2. Risk Factors of Uterine Disease in Dairy Cows

Cattle are more susceptible to uterine disease than other mammals (e.g., other ruminants) because bacterial contamination of the uterus in cattle dynamically alternates with clearance [8]. Most cows experience bacterial contamination of their uterus in the first two weeks after calving. Problems arise when this bacterial load becomes persistent. During and after parturition, the physical barriers of the female genital tract are compromised, thus creating the possibility of an ascending infection. Bacteria can get into the uterine lumen from the environment, as well as the skin and faeces of the animal, especially if hygienic conditions are not up to standard in the calving stable. Combined with the endometrial tissue damage and the immunosuppressed status of the fresh-calved cow (which can be worsened if the stress levels increase due to poor handling of the animal), this can easily lead to inflammation of various organs of the genital tract, especially of the uterus, and cause delayed involution [8].

Risk factors (Figure 1) for uterine disease include uterine damage caused or worsened by the following: stillbirth, twin calving, retained placenta, caesarean section; metabolic conditions like milk fever and ketosis; damage associated with displaced abomasum; a dysbalance between the immune system and the invading pathogens due to a disrupted leukocyte function; the type of uterine bacterial flora present [8,20]. Cows with a history of postpartum uterine disease and/or abnormal parturition have higher odds of developing subclinical uterine disease than cows without postpartum uterine infections and those with normal parturition previously. Therefore, multiparous cows have higher odds of being diagnosed with uterine disease than primiparous ones, but it is their parturition history, not the actual number of parturitions, that increases the risk [21,22]. There is a difference between the efficacy of the defence mechanisms of primiparous versus multiparous cows. Older cows have lower uterine elasticity and slower involution than younger ones. On the other hand, young animals’ limited exposure to bacterial contamination delays proper immune response, while older cows may eliminate pathogenic bacteria faster [18].

The most significant risk factor for developing metritis is a history of retained placenta. Consequently, all factors that increase the probability of retained placenta simultaneously increase the risk of developing metritis. A negative energy balance in the peripartal period and certain changes in body condition score also affect the odds of developing a uterine disease. A decrease in prepartum body condition score of at least 1 until days 28–35 postpartum significantly increases the risk. Once metritis occurs, the odds of ovarian inactivity on days 28–35 increase and 100-day milk production decreases [23,24]. Further significant risk factors for metritis in dairy herds are first pregnancy, calving in winter, having a male calf and a shorter gestation length [25].

Although endometritis in dairy cattle tends to be cured without intervention, a study found that approximately 25% of cows with endometritis had persistent or recurrent inflammation even after the voluntary waiting period [26]. Risk factors for endometritis, in addition to the above-mentioned, included summer calving, male offspring, calving assistance, induced calving, metritis, lameness, clinical mastitis and clinically relevant urovagina [26,27]. On the contrary, another study found that calving between November and April increased the risk of uterine infection in the first month postpartum [28]. These results can be explained by the different climatic conditions between studies. Dystocia increases the risk of endometritis both directly and indirectly (increasing the probability of metritis). There is a positive correlation between the degree of metritis and the odds of developing endometritis [18]. One research group found that the cleanliness of the animal or the environment was not a significant risk factor [27]. On the contrary, a different study concluded that the odds of developing uterine disease were not influenced by the gender and viability of the calf or calving assistance [23].

A herd-level risk factor for subclinical endometritis is housing freshly calved cows in bedded pack barns. Cow-level risk factors for subclinical endometritis include ketosis, acute metritis and higher milk production in primiparous cows. In contrast, multiparous cows with higher milk production showed lower odds of having subclinical endometritis in a study [29].

Risk factors for pyometra include hormonal dysfunction (early ovulation after calving combined with a retained corpus luteum) due to or combined with persistent infection of the uterus during and after parturition (previously explained in detail as risk factors for uterine disease in general) [8,11].

### 2.3. Diagnostic Methods

There are several diagnostic methods to detect the different forms of uterine disease. Szenci et al. [17] listed rectal palpation, transrectal ultrasonography, cytological examination, histopathological examination of endometrial biopsy samples, vaginal palpation, vaginoscopy and Metricheck™ (Simcro Tech Ltd., Hamilton, New Zealand). The metritis score system is an easy cow-side method, especially for puerperal metritis, that guides veterinarians on whether an animal requires treatment. Rectal temperature, decreased milk yield, dehydration, rumination and vaginal discharge are scored from 0 to 3 (0 being normal and 3 being the most severe symptom) [9,16]. Vaginoscopy, cytological and even ultrasonographic examinations have limited use in large dairy herds on a daily basis due to their time and equipment requirements. Endometrial biopsy sample histology has been considered the reference test for diagnosing endometritis. Similar results can be obtained with both the cytobrush technique and the low-volume uterine lavage [30]. McDougall et al. [31] concluded that the Metricheck™ device, which allows for the easy and precise sampling of vaginal discharge, had a higher sensitivity and lower specificity in diagnosing endometritis than vaginoscopy. They speculated that the diagnostic method that was first used might have stimulated uterine contractions and the second method increased the chances of detecting purulent discharge in the same cow. Pleticha et al. [32] used vaginoscopy as a reference method, in comparison to vaginal palpation and the Metricheck™ device, in three different groups of cows to avoid bias by vaginal stimulation. Significantly more cows were diagnosed with endometritis using the Metricheck™ device than using vaginoscopy or manual examination, but this did not result in a higher reproductive performance in the Metricheck group. Dubuc et al. [30] suggested that, in many cases, cytological and clinical endometritis may represent different manifestations of inflammation because only 36–38% of cows with clinical endometritis showed evidence of cytological endometritis. In a later study, Denis-Robichaud and Dubuc [33] reported this proportion as being 55% of the total. They suggested the introduction of a new term, purulent vaginal discharge (PVD), as it could indicate not only endometritis but cervicitis and vaginitis as well. Another study [34] described endometrial cytology, performed using the cytobrush technique or low-volume uterine flushing, as a minimally invasive method, which had no adverse effects on subsequent reproductive performance. They also suggested that the terms PVD and cytological endometritis should be used instead of the classical terminology [34]. Kelly et al. [35] also used PVD evaluation with the Metricheck™ device and ultrasonographic endometritis scoring and found both methods practical and reliable, while, in some instances, they could also detect different manifestations of reproductive tract diseases.

A Chinese study by Sun et al. [36] compared a multiplex PCR diagnostic method with clinical examination, a sulphur-containing amino acid test and conventional culture and biochemical tests from vaginal discharge samples. They tested the method with three major agents of endometritis in dairy cows: *Staphylococcus aureus*, *Escherichia coli* and *Bacillus cereus*. These three tested field isolates showed positive amplification, whereas other strains (*Pasteurella* spp., *E. faecalis*, *Streptococcus* spp.) showed no amplification. The study concluded that the multiplex PCR method had equal specificity and higher sensitivity than the conventional PCR method. Furthermore, it was cheaper and time-saving, as various pathogens could be detected simultaneously [36]. Another experiment [37] evaluated urinary test strips on uterine lavage samples. Leukocyte esterase, protein and pH were measured; however, the reagent strip, as an alternative cow-side method, had a relatively poor performance compared to endometrial cytology [37]. Lima [38] suggested potential new technologies to improve the diagnosis of uterine diseases. Activity monitors, biomarkers, immune cell profiling and machine learning algorithms are listed as promising additional diagnostic tools that can help to detect high-risk animals and predict the success rate of antimicrobial therapy.

Subclinical endometritis can be diagnosed by histological examinations of the uterus as a gold standard method in the absence of discharge. Cells can be collected by uterine lavage, cytobrush or cytotape techniques [39,40,41]. Cytobrush and uterine lavage are viable and comparable sampling methods [42,43]. Van Schyndel et al. [43] also tested leukocyte esterase strips, which proved to be an easy cow-side diagnostic method, either alone or in combination with other techniques, when the cut-point was set at leukocyte esterase (LE) ≥ 2. On the contrary, brix refractometry had a very poor performance in diagnosing subclinical endometritis [43]. Baranski et al. [44] collected endometrial surface scrapings by cytobrush and performed bacteriologic and cytologic examinations. They concluded that subclinical endometritis may be more associated with the recovery of the endometrium after calving than a bacterial infection [44]. McDougall et al. [45] had similar findings: they stated that the more important determinant of subsequent reproductive performance was inflammation, rather than the presence of pathogens.

Pyometra can be diagnosed by transrectal palpation of the uterus and/or transrectal ultrasonography by scanning uterine content and a persisting corpus luteum on one of the ovaries [42,46].

## 3. Bacterial Flora of the Reproductive Tract

Vaginal microbiota (microbiota in general include bacteria, viruses, fungi, protozoa and small parasites) is an essential element of reproductive health in dairy cattle, which changes over time due to internal (e.g., oestrus cycle; immune system) and external (e.g., infection; injury) factors. Most studies focused on the bacterial communities of the vagina and uterus [47]. Otero et al. [48] studied a group of heifers over the period of 18 months before their first insemination. The predominant bacteria of the vagina were *Enterococci* and *Staphylococci*, followed by a lower abundance of *Enterobacteriaceae* and *Lactobacilli*. The relative abundance of *Lactobacilli* seemed to increase with the age of the animal [48]. Quereda et al. [49] also suggested that—contrary to humans—the vaginal microbiota of healthy dairy heifers was not dominated by *Lactobacillus* spp. The most abundant genera or families, according to them, were *Ureaplasma*, *Histophilus*, *Corynebacteriaceae*, *Porphyromonas*, *Mycoplasma* and *Ruminococcaceae* [49].

Ault et al. [50] studied healthy multiparous beef cattle before insemination, and they detected 34 phyla and 792 genera. They found that the abundance of *Corynebacteria* and *Staphylococci* is significantly higher two days before insemination in cows that did not conceive [50]. They also described some changes in microbiota before insemination, indicating that the number of bacterial species in the reproductive tract significantly decreased over time, but this was not linked to pregnancy status [51].

Westermann et al. [52] sampled the uterus of 230 cows with vaginal discharge by a cytobrush technique, and in 19.6% of these samples, they did not find any bacteria. This result suggests that evaluating vaginal discharge alone comes with a fair share of false-positive diagnoses that lead to unnecessary or inadequate treatment. From the positive samples, they isolated *T. pyogenes*, *E. coli*, coagulase-negative *Staphylococci*, α-haemolytic *Streptococci*, *C. bovis* and *Bacillus* spp. [52]. Udhayavel et al. [53] found 16.66% of the samples collected from cows with clinical endometritis to be sterile. *E. coli*, *Klebsiella* spp., *Proteus* spp., *Pseudomonas aeruginosa* and *Clostridium* spp. were identified from endometritic samples. Based on in vitro antibiotic sensitivity testing, they found ceftriaxone to be effective in 64% of bacterial uterine infection cases, followed by gentamicin, enrofloxacin and chlortetracycline, in 32% of cases [53].

Knudsen et al. [54] suggested that examining uterine flush samples alone might not provide a complete picture, as the microbiota of the endometrium were absent. They compared uterine flush samples with endometrial biopsies and found a correlation in the microbiota, but they also stated that endometrial biopsy samples were more diverse. They concluded that *Porphyromonadaceae*, *Fusobacteriaceae*, *Leptotrichiaceae* and *Mycoplasmataceae* may be associated with uterine disease—all of them, however, could also be isolated from healthy cows, while they observed *Ruminococcaceae* in high abundance in both healthy and endometritic cows [54].

A Brazilian research group [55] obtained 205 bacterial and 120 yeast isolates from the vagina of 20 beef cows. The most frequent bacteria from their isolates were *Staphylococcus* spp. and *E. coli*. Yeast colonies were identified as *Candida tropicalis*, *C. albicans* and *C. krusei*. Their *Staphylococcus* spp. bacterial isolates were tested for the antimicrobial sensitivity of 16 drugs. The lowest sensitivity was presented to tetracycline and erythromycin (46.9%), followed by amoxicillin (65.6%) and rifampicin (71.8%)—all other drugs were 100% effective in the study [55]. Another Brazilian group [56] also collected vaginal samples from 20 healthy beef cattle, and they found that *Firmicutes*, *Bacteroides* and *Proteobacteria* were the three most abundant phyla. They also reported many unclassified bacteria, which indicates that further research is necessary to fully understand bovine reproductive tract microbiota. Eight out of nine genera that they identified could also be found in the bovine gastrointestinal tract or faeces, and only *Aeribacillus* (*Firmicutes*) (the most abundant) has not yet been isolated from the gastrointestinal tract. They could not find any association between specific vaginal microbiota and pregnancy status or parity. Regarding the detection of fungi, authors isolated the phyla *Ascomycota* and *Basidiomycota*, with the *Ascomycota* population decreasing after conception [56]. These results are consistent with the findings of Chen et al. [57], who also described *Firmicutes*, *Proteobacteria* and *Bacteroidetes* as the most abundant phyla. This research group tried to find differences between open and inseminated cows, but they could not detect any statistical difference [57]. Swartz et al. [58] had similar findings, describing *Bacteroidetes*, *Fusobacteria* and *Proteobacteria* as the predominant bacterial phyla.

The vaginal and uterine microbiome of a cow changes dynamically after parturition. Healthy cows develop differentiated uterine flora early on, while cows later diagnosed with uterine diseases tend to show a loss of bacterial diversity and dominance by a few bacterial taxa [59]. Jeon et al. [60] suggested that on day 2 postpartum, *Bacteroidetes* spp. and *Fusobacterium* spp. are already more abundant in pre-metritic cows and between days 4 and 8, this tendency becomes even more apparent. On the other hand, Tasara et al. [61] only found statistically significant changes in the species richness and alpha diversity of uterine microbial communities between healthy and metritic cows between days 7 and 10 postpartum. The synergistic effect was also supported by Wang et al. [62] and Bicalho et al. [63], who found an increased abundance of *Fusobacterium* spp. [62,63] and the unique presence of *Trueperella* spp. [62,63] and *Peptoniphilus* spp. [62] on day 30 and between days 25 and 35 postpartum in cows with clinical endometritis, respectively. However, in subclinical endometritis 30 days postpartum, almost none of the known intrauterine pathogens were detected, and uterine flush samples were characterized by the enrichment of *Lactobacillus* spp. and *Acinetobacter* spp. [62]. Similar results were obtained on day 35 postpartum by Pascottini et al. [64], who reported that cows with clinical endometritis showed a decreased bacterial diversity, with *Bacteroidetes* spp. and *Fusobacterium* spp. being more dominant, whereas cows with subclinical endometritis had similar microbiota to healthy cows.

An interesting contrast to the uterine characteristics described above appears in the study of Wang et al. [65] regarding vaginal flora: healthy cows had a significantly lower vaginal bacterial diversity compared to cows with endometritis, and diseased cows lacked dominant bacterial species [65]. However, other studies suggest *E. coli* [66], *Histophilus* spp. [67,68], *Mogibacterium* spp. [68], *Bacteroides* spp. [67,69], *Fusobacterium* spp. [69] and *Proteobacterium* spp. [69] are more abundant in the vaginal microbiome of diseased cows. Parity seems to affect the composition of vaginal microbiota in such a way that multiparous cows have a significantly greater diversity [70], as well as having greater diversity in their uterine microbiota [71].

## 4. Disease Prevention

Placing the emphasis on disease prevention rather than treatment is beneficial to the cow’s health and the farm’s economy. A housing design that allows cows to separate from the herd during calving, a spacious calving pen and a clean environment help to avoid infection during and after parturition and reduce stress. An optimal diet throughout pregnancy and early lactation prevents metabolic disorders, deficiencies and suboptimal body conditions. Educating staff on calving assistance and identifying high-risk animals in advance can reduce risk factors that arise during calving. Using female sexed semen and having smaller female calves compared to bigger males can prevent problems around calving too. Close monitoring in the early days of lactation and a quick response to any sign of disease prevent the escalation of health issues Figure 1, [72,73].

## 5. Treatment Options

Defeating endometritis (both clinical and subclinical) is not a medical emergency, but it is very important for the reproductive performance of the cow. It is important to note that spontaneous healing of endometritis may occur if oestrus successfully clears up the uterus. Metritis, on the other hand, may require systemic treatment to restore the general good condition of the cow [74]. Metritis treatment usually includes a combination of antibiotics, NSAIDs and hormones, including uterotonics. Generally speaking, the most commonly used antibiotics have been tetracycline, amoxicillin, ampicillin and sulfonamides, often with trimethoprim, cephapirin, ceftiofur and benzylpenicillin procaine. When deciding which antibiotic agent could be included in the treatment plan, one must consider legal restrictions, effectiveness against Gram negatives and anaerobes, the form and severity of disease, and non-antibiotic options [75]. The most frequently used NSAIDs are flunixin meglumine, ketoprofen, meloxicam and carprofen. Hormones and uterotonic drugs include oxytocin, which can be used in the first few hours (max. 72 h) after calving, and, after this timeframe, prostaglandin F2α can be used from day 3 postpartum [24,76]. The benefit of PGF2α is believed to be oestrus induction in the presence of a PGF2α responsive corpus luteum once the ovarian activity is restored during involution. Oestrus promotes the clearance of bacteria and inflammatory products [77]. Although Galvao et al. [77] found that treatment with PGF2α did not decrease the prevalence of subclinical endometritis at the time points they studied (days 35 and 49 postpartum), it did improve conception rate and the odds of pregnancy in cows with a low body condition score. However, other researchers disagree on whether PGF2α administration affects certain cases, e.g., before week 4 postpartum or in cows without a palpable corpus luteum [42]. Supportive therapy for metritis includes fluid therapy, calcium and energy supplementation [42]. The prognosis depends on the severity of symptoms [17].

There is an ongoing dispute about intrauterine versus systemic (antibiotic) treatments. There have been some promising results regarding intrauterine treatment with cephapirin [78], chlortetracycline [13] and dextrose [79]. A research group in India compared five different intrauterine treatment (antibiotic and/or non-antibiotic) methods to a control group and concluded that Lugol’s iodine, followed by *E. coli* LPS 24 h later, was the most effective in increasing first service pregnancy rate, although cervicovaginal mucus was found not clear in 40% of the cows in this treatment group [80]. However, some suggest that intrauterine treatments irritate the mucous membrane of the uterus, and active agents barely reach the deeper histological layers of the uterus and the rest of the reproductive organs; therefore, they provide no advantage compared to no-treatment control groups [17,24,81].

Haimerl et al. [82] concluded that a significant decline in metritis prevalence occurred following treatment with the most used antibiotic, ceftiofur. Ceftiofur has been approved for the treatment of puerperal metritis on five consecutive days in Europe and the USA (although legal regulations may differ among countries; therefore, it is impossible to make a statement on treatment practices that is valid globally), and it does not require milk withdrawal. With the emerging antimicrobial resistance of zoonotic organisms, there is a high demand for prudent antibiotic use in food animals worldwide. For each disease, including reproductive tract diseases, it is critically important to select the most appropriate drug at its optimal dosage and duration. This way, side effects and the pressure to select resistant strains can be minimized. Ceftiofur, the most frequently used drug to treat metritis, for example, is a third-generation cephalosporin, and as such, it is valued for treating severe to life-threatening human infections. Ceftiofur has been associated with developing resistance to ceftriaxone, which is only available for humans [83,84]. Haimerl and Heuwieser [83] even reported that ceftiofur treatment does not improve reproductive performance, although clinical improvement is evident. Uterine pathogens such as *E. coli* may contain different antibiotic resistance genes and might even be multi-drug-resistant, including ceftiofur [85].

There is a general need for comparative studies of different antibiotic and non-antibiotic treatment options. One rare example of that is the study of Jeon et al. [84], who compared ceftiofur to ampicillin and a no-treatment control group. They found that regardless of treatment, uterine microbiota became more homogenous over time after parturition but ceftiofur contributed to more dynamic changes from days 1 to 6. The relative abundance of *Bacteroidetes* increased significantly after ceftiofur administration, whereas it did not change after ampicillin treatment or in the absence of treatment. This indicates that *Bacteroidetes* seem more resistant to ceftiofur; therefore, new therapeutic methods with more effectivity against this phylum should improve the cure rates of metritis [84]. Another comparison of ceftiofur to ampicillin was carried out by Merenda et al. [86], who concluded that ceftiofur reduced rectal temperature but, in primiparous cows, reduced the pregnancy rate and increased the median days to pregnancy compared to ampicillin. Ampicillin treatment, on the other hand, resulted in a greater prevalence of purulent vaginal discharge and cytological endometritis compared to ceftiofur [86].

Lima et al. [87] compared the economic aspects of ceftiofur and ampicillin treatment options for cows without metritis. The analysis considered the cost of therapy, reproductive management, discarded milk and the income from saleable milk and culled cows. They concluded that the choice of antibiotic did not significantly alter survival rates, reproductive performance or the costs of the disease [87].

## 6. Treatment Alternatives with Regard to Probiotics

In recent years, promising experiments have begun to evaluate the therapeutic value of 50% dextrose solution and proteolytic enzyme (trypsin, chymotrypsin, papain) solutions as substitutes for antibiotics. Further examinations are required to confirm the positive effects of these agents, as results have been somewhat inconsistent so far [17,38,79,88]. Vaccination against the predominant bacteria has also been described as a promising alternative to antibiotics. Subcutaneous vaccinations containing inactivated bacterial components and/or protein subunits were found to significantly reduce the incidence of puerperal metritis in heifers, leading to improved reproductive performance [89]. Other areas of investigation include but are not limited to probiotics, bacteriophages, acetylsalicylic acid, botanical essential oils, chitosan microparticles and even acupuncture therapy, with varying success—further research is needed in this field [38,90,91,92].

The use of probiotics is a new and promising research area for the prevention or treatment of uterine disease. The definition of probiotics, coming from the Food and Agriculture Organization (FAO) and the World Health Organization (WHO) states that probiotics are “live microorganisms, which, when administered in adequate amounts, confer a health benefit on the host” [93]. This definition was reinforced by an expert panel with minor grammatical changes and was supplemented with additional recommendations [94]. It applies to humans and animals and accepts positive health effects outside the gastrointestinal tract but requires an adequate number of live microorganisms at the time of administration [93].

There are several mechanisms of probiotic actions (Figure 2), which include enhancing epithelial cell barrier functions by increasing the expression of genes involved in tight junction signalling, establishing biofilms on mucosal layers, immune modulation, competition for adhesion and preventing the adhesion of pathogen bacteria, competition for nutrients, the production of antimicrobial compounds (bacteriocins, organic acids, antimicrobial proteins, enzymes, H_2_O_2_ and CO_2_) and maintaining optimal vaginal pH [95].

So far, the group of bacteria that are most often studied as potential intravaginal probiotics is the group of lactic acid bacteria (LAB; Table 1). This is a diverse group of Gram-positive bacteria from different taxa that produce lactic acid as the primary endproduct of carbohydrate fermentation. Otero et al. [96] isolated 76 strains of *Lactobacillus* spp. and 7 strains of *Streptococcus* spp. from the vagina of healthy heifers and adult cows. After that, they performed in vitro experiments to evaluate the probiotic potential of these strains and found that the majority of them have the potential to inhibit *E. coli*, whereas only a few strains showed the potential to inhibit *T. pyogenes*. Two years later, they were able to isolate 82 *Lactobacillus* spp. strains from the bovine vagina and performed screening assays of antagonistic substance production and surface characteristics. Three strains showed the highest potential to be included in future studies. These are *Lactobacillus gasseri* CRL1412 and CRL1421 and *Lactobacillus delbrueckii* subsp. *delbrueckii* CRL1461 [97]. 

Genís et al. [98,99] tested four LAB species in vitro and concluded that, among them, the following three strains have the potential to inhibit *E. coli* infection: *Pediococcus acidilactici* (reduced *E. coli* infection by 89.7%), *Lactobacillus sakei* (decrease in infection by 87%) and *Lactobacillus reuteri* (decrease in infection by 73.5%). However, when infection was combined with inflammation, only *P. acidilactici* and *L. reuteri* showed the potential to reduce *E. coli* infection (up to 83%). After a separate strain experiment, LAB combinations were tested and a combination of *L. rhamnosus* ratio 25, *P. acidilactici* ratio 25 and *L. reuteri* ratio 2 proved to be the most effective in modulating *E. coli* infection (*E. coli* count was reduced by 95.1%) and endometrial inflammation (*E. coli* infection was reduced by 89.78%) in vitro [98,99]. Liu et al. [100] also tested *L. rhamnosus* against *E. coli* infection in vitro and concluded that *L. rhamnosus* pretreatment has the potential to limit the inflammatory response to *E. coli* and the subsequent damage to bovine endometrial epithelial cells.

*P. acidilactici* was the focus of the research of Wang et al. [101]. They could isolate *Enterococcus*, *Lactobacillus* and *Pediococcus* from both healthy and metritic cows and reported that the bacterial load of the vaginal mucus increased after calving. *E. coli* was the predominant species in the vaginal microbiota of cows with metritis, and they used different strains as indicators for an inhibition assay. Two isolates of *P. acidilactici* (FUA3138 and FUA3140) produced a bacteriocin, namely, the pediocin AcH/PA-1. *P. acidilactici* FUA3072 was used as a reference strain as it had been previously characterized [101].

Rodríguez et al. [102] characterized the lactobacilli microflora of dairy and beef cows and identified the facultative heterofermentative group as the most dominant. *L. plantarum* was the most abundant species. *L. acidophilus* was the most dominant species of the obligate homofermentative group. The third group, the one of the obligate heterofermentatives, was only presented by *L. brevis* [102]. Niu et al. [103] examined *Lactobacillus* strain SQ0048 and described the genes and pathways involved in the adhesion to host cells and the inhibition of pathogens, including the Interleukin-17 (IL-17) signalling pathway and the antigen processing and presentation pathways. 

Following the in vitro experiments, a few in vivo tests were performed (Table 2). One research group [104] used a combination of three bacterial strains—*Lactobacillus sakei* FUA 3089, *P. acidilactici* FUA 3140 and FUA 3138—intravaginally in dairy cows once a week from 2 weeks before calving until 4 weeks following calving [104]. The LAB mixture reduced the occurrence of purulent vaginal discharge at 3 weeks after calving and the plasma haptoglobin concentrations at 2 and 3 weeks after calving. The treated multiparous cows also had a higher milk yield than the control group, but no difference could be measured in primiparous cows [104]. The same LAB combination was used in another piece of research at weeks -2 and -1 pre-calving in one treatment group and at weeks -2 and -1 pre-calving, plus week 1 after calving, in a second treatment group. The results suggest that intravaginal LAB treatment can reduce the incidence of metritis and the serum concentration of lipopolysaccharide-binding proteins [105]. Cows in both treatment groups had smaller gravid uterine horns and body sizes at day 14 after calving compared to the control group. The first treatment group had fewer days open, but the second group did not differ from the control group [106]. This experimental setup was also used to evaluate the effect of intravaginal LAB treatment on milk production. The results showed that multiparous cows in both treatment groups had higher milk production and feed efficiency compared to the control group, similar to another study [104], and primiparous cows in the second treatment group also had a higher milk yield than the control group [107].

Other studies found that two intravaginal doses of LAB mixture—*L. rhamnosus* CECT 278, *P. acidilactici* CECT 5915 and *L. reuteri* DSM 20016—per week, starting from week -3 pre-calving, reduced metritis prevalence, whereas an intrauterine treatment on day 1 postpartum did not have such an effect. Both treatment types reduced blood neutrophil gene expression [108,109]. Yang et al. [110] used lactic acid bacteria and stated that their treatment could be an alternative to antibiotics, as most sick cows in the study returned to normal physiological status after the lactic acid bacteria treatment. *Lactobacillus* species have three fundamental mechanisms against pathogens. They prevent the initial adhesion of pathogens to epithelial cells. They maintain a normal pH level in the vagina, which inhibits the reproduction of pathogenic bacteria. *Lactobacillus* species can also produce antibacterial substances that can directly or indirectly destroy pathogens [110]. García-Galán et al. [111] also described in vitro the pH acidifier role of *Lactobacillus* spp. in the bovine vaginal mucus, although a sufficient concentration of the probiotic bacteria is required to achieve significant growth. Peter et al. [112] showed that the administration of *Lactobacillus buchneri* in utero, even only once, could improve the reproductive performance of healthy cows and cows with subclinical endometritis. This was supposed to be due to the initial stimulatory effect of *L. buchneri* on the local immune system and defence mechanisms [112]. More recent in vivo studies also reported lowered incidence rates of uterine infections, improved uterine involution, increased fertility, a reduced number of oestrus induction days and increased conception rate in buffalo [113] or cows [114,115].

**Table 2 vetsci-11-00066-t002:** The different probiotic strains used in in vivo studies for treating the reproductive tract and their effects.

Strains Used	Species	Target	Effects	References
*Lactobacillus sakei* FUA 3089 *Pediococcus acidilactici* FUA 3140 *P. acidilactici* FUA 3138	cattle	vagina, before and after calving	Lowered incidence of uterine infections and purulent vaginal discharge, and improved local and systemic immune responses. Multiparous cows had greater milk production and feed efficiency. The concentration of plasma haptoglobin was lower. Increased concentrations of serum progesterone level and earlier cyclicity of ovaries.	[104,105,106,107]
*L. rhamnosus* CECT 278 *P. acidilactici* CECT 5915 *L. reuteri* DSM 20016	cattle	vagina, before calving	Reduced metritis prevalence	[108]
*L. rhamnosus* CECT 278 *P. acidilactici* CECT 5915 *L. reuteri* DSM 20016	cattle	vagina, before calving	LAB decreases the amount of E. coli in the endometrium ex vivo.	[109]
*L. buchneri* DSM 32407	cattle	uterus, lactating cows on d 24–30 postpartum	Stimulatory effect on the local immune system. A higher proportion of cows were pregnant after the first service. The endometrial mRNA expression of several pro-inflammatory factors was lower.	[112]
*Lactiplantibacillus plantarum* KUGBRC *P. pentosaceus* GBRCKU	buffalo	vagina, after calving, with clinical endometritis	Reduced number of oestrus induction days and lower incidence of endometritis.	[113]
*L. rhamnosus* *P. acidilactici* *L. reuteri*	cattle	vagina, before calving	Decreased incidence of metritis and increased conception rate in multiparous cows.	[114]
*L. farraginis* NRIC 0676 *L. rhamnosus* NBRC 3425	cattle	vagina, before and after calving	Lowered incidence rates of uterine infections, improved uterine involution and increased fertility	[115]

Styková et al. [116] isolated five strains (*L. büchneri* 5K and 24S8, *L. mucosae* 29S8, 9/K and 5/Kb) that could be promising candidates for in vivo testing as reproductive tract probiotics. These strains produced hydrogen peroxide and lactic acid and adhered to the vaginal mucus (except for *L. büchneri* 24S8). They showed an inhibition zone against three uterine pathogens: *T. pyogenes*, followed by *F. necrophorum* and *Gardnerella vaginalis* [116].

The survivability of bacteria in the vagina and the uterus depends on multiple factors, including tolerance for physiological pH levels. Clemmons et al. [117] examined 30 cows and measured vaginal pH levels between 6.15 and 7.44, with a mean of 6.69 ± 0.14. Uterine pH levels ranged from 5.62 to 6.52, with a mean of 6.06 ± 0.09. Swartz et al. [58] also measured vaginal pH of 20 cows and described a range from 6.5 to 8.7, with a mean of 7.3 ± 0.63. On a larger sample size of Holstein-Friesian cows, Beckwith-Cohen et al. [118] found no correlation between vaginal pH and days in milk. While cows maintained a median pH value of 7.50, heifers showed a significant increase, starting at a median pH value of 7.25 before calving, then reaching a median of 7.75 during the first week postpartum, before settling at a median of 7.50 [118].

Further in vivo studies are required to provide extensive knowledge about potential intravaginal probiotic bacterial strains and their optimal application under field conditions. The clinical safety and efficacy of probiotics, focusing on the pharmacological and toxicological aspects, is crucial. Once a particular bacterial strain is chosen, it must undergo different mandatory evaluations, such as safety and innocuity assays and functional and technological characterization, before it can be incorporated into a veterinary medical product. Evaluation from the economic point of view is also unavoidable. Further target animal studies are required to provide extensive knowledge about the potential intravaginal probiotic bacterial strains and their optimal application under field conditions. The major steps in intravaginal probiotic product development are displayed in Figure 3.

Another main concern is antibiotic resistance. Probiotic strains cannot contain transmissible genes that encode resistance to medically important antibiotics or are related to virulence factors. Selected microorganisms must be evaluated based on their ability to adapt to the vaginal ecosystem and be economically included in a pharmaceutical product [119,120].

## 7. Conclusions

Uterine disease is still a significant health issue on most commercial dairy cattle farms, although it has been investigated in detail, especially following the early 2000s. There are a number of predisposing factors, but not all of them, can be entirely eliminated on a farm, which means that uterine disease cannot be prevented by management tools only. Commonly applied treatment protocols require manual labour, extensive drug usage and repeated follow-up examinations. With the global spread of antibiotic resistance, there is a growing demand for developing alternative treatment options and focusing more on prevention in farm animal medicine. In the case of cattle uterine disease, the most promising new area of research is probiotics. Probiotics have already been successfully used orally in cows. However, in vivo studies involving a more significant number of cows need to be conucted to evaluate the potential of probiotics for uterine disease prevention.

## Figures and Tables

**Figure 1 vetsci-11-00066-f001:**
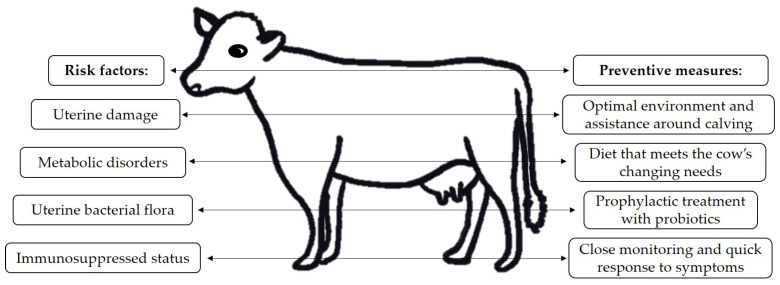
The most important risk factors for cattle uterine disease and which measures can help prevent them.

**Figure 2 vetsci-11-00066-f002:**
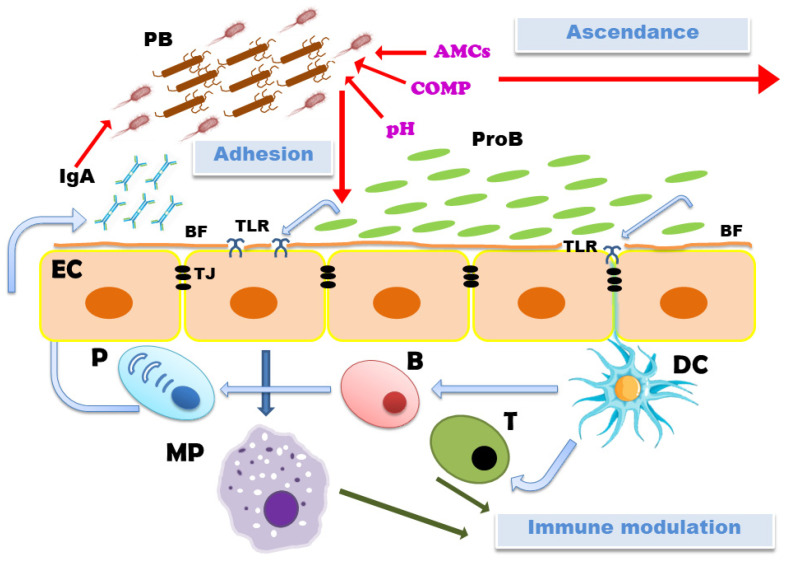
Mechanisms of action of probiotic bacteria (**ProB**) that may decrease the possibility of post-puerperal diseases in cows involve direct negative action on pathogen bacteria (**PB**); by the production of the biofilms (**BF**) on the surface of epithelial cells (**EC**), competing for essential nutrients (**COMP**), production of antimicrobial compounds (**AMCs**) and maintaining the optimal vaginal **pH**. The mode of action for probiotic bacteria also includes enhancing epithelial cell barrier functions by increasing the expression of genes involved in the tight junction (**TJ**) signalling and modulation of the immune system via the activation of toll-like receptors (**TLR**). Further abbreviations used in the figure: **DC**: dendritic cells; **T**: T-lymphocytes; **B**: B-lymphocytes; **P**: plasma cells; **MP**: macrophages. Red arrows represent inhibitory and green arrows represent stimulatory action on the highlighted processes.

**Figure 3 vetsci-11-00066-f003:**
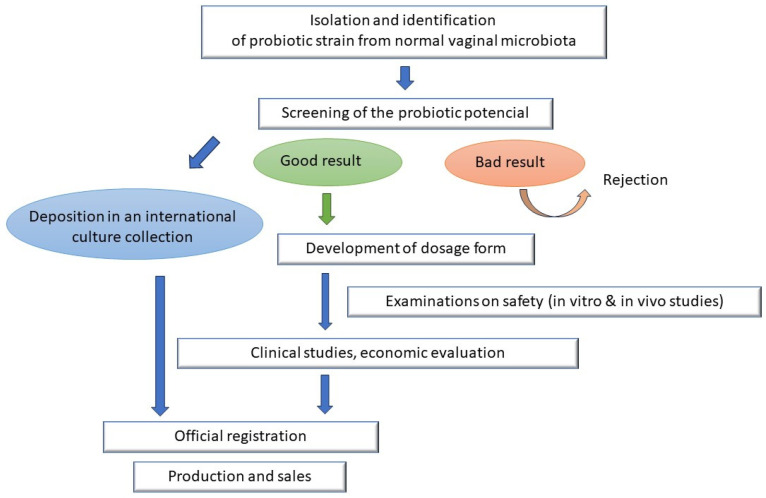
Flowchart of intravaginal probiotic product development.

**Table 1 vetsci-11-00066-t001:** The different probiotic strains used in in vitro studies.

Strains Used	Species	Source	Effects	References
76 *Lactobacillus* spp. strains 7 *Streptococcus* spp. strains	cattle	Vagina	Although most strains were able to inhibit *E. coli* due to their acid production, only a few strains were able to inhibit *T. pyogenes*.	[96]
82 *Lactobacillus* strains; *Lactobacillus fermentum* *L. gasseri* CRL1412, *L. gasseri* CRL1421, *L. delbrueckii* subsp. *delbrueckii* CRL1461	cattle	vaginal swab	95% produced H_2_O_2_; more than 80% produced lactic acid; no strains were bacteriocin producers. Listed isolates were able to inhibit *E. coli* or *T. pyogenes*.	[97]
*Pediococcus acidilactici* CECT 5915, *L. sakei* DSM 20100, *L. reuteri* DSM 20016, *L. rhamnosus* CECT 278	cattle	obtained from bacterium strain collections	Inhibition of infection with *E. coli* when inflammation is present. The combination of *L. rhamnosus* ratio 25, *P. acidilactici* ratio 25 and *L. reuteri* ratio 2 was most effective in reducing E. coli infection with or without basal tissue inflammation compared to single LAB strains.	[98,99]
*Lactobacillus rhamnosus* GR-1, ATCC 55826	cattle	obtained from bacterium strain collections	Limits the inflammatory response to *E. coli* infection and subsequent endometrial epithelial cell damage.	[100]
*Pediococcus acidilactici* (FUA3138 and FUA3140)	cattle	vaginal swab	Production of bacteriocin against *E. coli*.	[101]
*Lactobacillus* spp. (mainly *L. plantarum*, and *L. rhamnosus*, *L. curvatus*, *L. delbrueckii delbrueckii*, *L. acidophilus*)	cattle	vaginal flush	13 of 29 strains were characterized by H_2_O_2_ production	[102]
*Lactobacillus* strain SQ0048	cattle	vagina	Adhesion to host cells (endoplasmic reticulum protein processing pathways); Interleukin-17 signalling pathway involved in pathogen inhibition, antigen processing and presentation pathways.	[103]

## Data Availability

All data are contained within the article.

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
