# Peer review of "Uterine Disease in Dairy Cows: A Comprehensive Review Highlighting New Research Areas"

_vetsci, 2024, doi:10.3390/vetsci11020066_

Round 1

Reviewer 1 Report

Comments and Suggestions for Authors

This review from Zsóka Várhidi briefly summarized the recent knowledge on postpartum bovine uterine infection, mainly metritis and endometritis, with a particular emphasis on the treatment potential of probiotics. The manuscript is informative and well written. It would be better to add another section with regard to the disease prevention. Minor suggestions are listed below:

1. “2.2. Risk factors of uterine disease in dairy cows”. Environmental (hygiene) and management (stress) issue can be added.

2. Figure 1. Please add a more detailed description in figure legend.

3. line 403/405, “have the potential to inhibit...”/”showed potential to reduce...”. What does potential indicate? Please describe it in more detail objectively.

4. line 405/410, E.coli (italics).

5. line 427/463 (throughout the article), in-vivo or in vivo? Please be consistent throughout the manuscript.

Author Response

Reviewer 1.
This review from Zsóka Várhidi briefly summarized the recent knowledge on postpartum bovine uterine infection, mainly metritis and endometritis, with a particular emphasis on the treatment potential of probiotics. The manuscript is informative and well written. It would be better to add another section with regard to the disease prevention. Minor suggestions are listed below:

AU: Thank you for your effort to read our manuscript, and for your valuable comments and suggestions. We tried to do our best to improve the overall quality of the manuscript. Since the reviewers had several suggestions, we did not use the track changes function. Instead, the changed parts are indicated in yellow. We added a section about the disease prevention.

1. “2.2. Risk factors of uterine disease in dairy cows”. Environmental (hygiene) and management (stress) issue can be added.

AU: Thank you, it is corrected.

2. Figure 1. Please add a more detailed description in figure legend.

AU. A more detailed description is added in the figure legend.

3. line 403/405, “have the potential to inhibit...”/”showed potential to reduce...”. What does potential indicate? Please describe it in more detail objectively. 

AU: It is now corrected.

4. line 405/410, E.coli (italics).

AU: We carefully checked the text and tried to write all Latin names in italics.

5. line 427/463 (throughout the article), in-vivo or in vivo? Please be consistent throughout the manuscript.

AU: It is now corrected throughout the text.

Reviewer 2 Report

Comments and Suggestions for Authors

Thank you for your efforts in this review. However, there is not sufficient scientific merit for publication. I recommend that this be submitted to a veterinary education journal.

Comments on the Quality of English Language

The use of English is fine.

Author Response

Reviewer 2.

Thank you for your efforts in this review. However, there is not sufficient scientific merit for publication. I recommend that this be submitted to a veterinary education journal.

AU: Thank you effort to read our manuscript and for your suggestion. We felt the topic fits VetSci., but we can change our intentions if the editors say so.

Reviewer 3 Report

Comments and Suggestions for Authors

This review (vetsci-2805364: Uterine disease in dairy cows – a comprehensive review highlighting new research areas) was described about the recent knowledge on the definition, risk factors, diagnostic methods, and treatment of uterine diseases in dairy cows. This review was comprehensively well written, especially the alternative treatment (probiotics treatment) for uterine diseases. We think that some modifications would make the content more readable. Please confirm my comments.

General comment.

1. You should add the explanation of the risk factors of Pyometra in the section of "2.2. Risk factors of uterine disease in dairy cows".

2. L126-128; This information  were similar to the information of above. I think that it is better to explain the risk factors for the causes of endometritis separately for clinical and subclinical endometritis. I think that explaining each separately will reduce repetition of explanations.

Minor comments

Please describe the reference for this information.

1. L86-89

2. L103-104

3. L111-113

4. L118-122

5. L122-123

6. L134-135

7. L135-136

Author Response

Reviewer 3.

This review (vetsci-2805364: Uterine disease in dairy cows – a comprehensive review highlighting new research areas) was described about the recent knowledge on the definition, risk factors, diagnostic methods, and treatment of uterine diseases in dairy cows. This review was comprehensively well written, especially the alternative treatment (probiotics treatment) for uterine diseases. We think that some modifications would make the content more readable. Please confirm my comments.
AU: Thank you for your effort to read our manuscript, and for your valuable comments and suggestions. We tried to do our best to improve the overall quality of the manuscript. Since the reviewers had several suggestions, we did not use the track changes function. Instead, the changed parts are indicated in yellow.
General comment.
1. You should add the explanation of the risk factors of Pyometra in the section of "2.2. Risk factors of uterine disease in dairy cows".
AU: Thank you. These risks are now explained in the desired section.
2. L126-128; This information  were similar to the information of above. I think that it is better to explain the risk factors for the causes of endometritis separately for clinical and subclinical endometritis. I think that explaining each separately will reduce repetition of explanations. 
AU: We rephrased the text for better clarity and removed the repetitions wherever possible.
Minor comments
Please describe the reference for this information. 
1. L86-89
AU: References 16 and 17 are added to the end of the next sentence.
2. L103-104
AU: This part was removed from the current version.
3. L111-113
AU: It is corrected.
4. L118-122
AU: This part was removed from the current version.
5. L122-123
This part was removed from the current version.
6. L134-135
AU: This whole paragraph belongs to one reference mentioned at the end.
7. L135-136
AU: This whole paragraph belongs to one reference mentioned at the end.

Reviewer 4 Report

Comments and Suggestions for Authors

Although the paper adds nothing new to the field (many reviews have already been written on this topic, and there are numerous research papers in the literature), it is well written and covers a large and complex subject.

However, some parts (especially paragraph 2 “Uterine disease of dairy cows”) are written in a didactic style and lack a ‘scientific research’ approach.

Some bibliographical references are in Hungarian (N° 2, 12, 19, 34), I would suggest adding at least references in English as well.

Specific comments

-       Line 50 “In Grade 1, or acute puerperal metritis, there is no sign of systemic illness or fever”: please check this definition. According to Sheldon et al. (2006) “Puerperal metritis is an acute systemic illness” and for Credille et al. 2014 “Acute puerperal metritis (APM), defined as the presence of a fetid, watery uterine discharge, an enlarged, flaccid uterus, and overt signs of systemic illness…”

-       Citation 5: could not find it.

-       Line 70: I think that ref. 11 is not relevant.

-       2.2. Risk factors of uterine disease in dairy cows: the same concepts are often repeated, I would make this paragraph a little more structured.

-       Line 117: ref. 13 is a review, I will cite the study it refers to directly.

-       Line 163: please check, I think the percentages are 36-38%

-       Line 245: perhaps I would point out at the beginning of the paragraph that the study concerns the vaginal microbiota.

-       Line 247: I think that the reported sensitivity refers to the Staphylococcus isolates. Maybe it would be better to rewrite this part to make it clearer.

-       Line 259 “…with Ascomycota showing a tendency to reduce the odds of pregnancy”: I am not sure of this. The authors say “Ascomycota dominates all groups, showing a tendency to reduction in pregnancy condition”: they did not mention the odds of pregnancy. Am I wrong about that?

-       Lines 268 and 272: ref. 52 is a review, I would suggest citing the researches directly. Also check ref. 66 in line 272: according to the sentence, the authors contradict themselves.

-       Line 278: please check, I think Wang et al. found Peptoniphilus, not Bicalho et al.

-       Line 283: ref. 69 is not correct, please leave just ref. 68.

-       Line 292: ref. 72 is not correct.

-       Line 315: Treatment with PGF2alpha did not affect prevalence of SCE either at 35 or at 49 DIM (21 DIM not mentioned).

-       Line 364, ref. 85: I believe this work deals with antibiotic therapy and not with alternative treatment methods.

-       Line 449: blood neutrophil gene expression.

-       Line 484: I think the reference is wrong, it is 117 and not 118, please check.

Author Response

Reviewer 4.
Although the paper adds nothing new to the field (many reviews have already been written on this topic, and there are numerous research papers in the literature), it is well written and covers a large and complex subject.
However, some parts (especially paragraph 2 “Uterine disease of dairy cows”) are written in a didactic style and lack a ‘scientific research’ approach.
AU: Thank you for your effort to read our manuscript, and for your valuable comments and suggestions. We tried to do our best to improve the overall quality of the manuscript. Since the reviewers had several suggestions, we did not use the track changes function. Instead, the changed parts are indicated in yellow.
Some bibliographical references are in Hungarian (N° 2, 12, 19, 34), I would suggest adding at least references in English as well. 
AU: We added English citations to support each Hungarian references.

Specific comments
-       Line 50 “In Grade 1, or acute puerperal metritis, there is no sign of systemic illness or fever”: please check this definition. According to Sheldon et al. (2006) “Puerperal metritis is an acute systemic illness” and for Credille et al. 2014 “Acute puerperal metritis (APM), defined as the presence of a fetid, watery uterine discharge, an enlarged, flaccid uterus, and overt signs of systemic illness…
AU: It is now corrected, and another citation was added.
-       Citation 5: could not find it.
AU: A lot was changed in reference numbers, but we rechecked each.
-       Line 70: I think that ref. 11 is not relevant.
AU: The reference is removed.
-       2.2. Risk factors of uterine disease in dairy cows: the same concepts are often repeated, I would make this paragraph a little more structured. 
AU: We rephrased the whole section for better clarity, and removed repetitions wherever possible.
-       Line 117: ref. 13 is a review, I will cite the study it refers to directly.
AU: It is corrected.
-       Line 163: please check, I think the percentages are 36-38% 
AU: It is corrected.
-       Line 245: perhaps I would point out at the beginning of the paragraph that the study concerns the vaginal microbiota.
AU: It is corrected.
-       Line 247: I think that the reported sensitivity refers to the Staphylococcus isolates. Maybe it would be better to rewrite this part to make it clearer. 
AU: It is now corrected.
-       Line 259 “…with Ascomycota showing a tendency to reduce the odds of pregnancy”: I am not sure of this. The authors say “Ascomycota dominates all groups, showing a tendency to reduction in pregnancy condition”: they did not mention the odds of pregnancy. Am I wrong about that? 
AU: Indeed. Thank you, it is now corrected.
-       Lines 268 and 272: ref. 52 is a review, I would suggest citing the researches directly. Also check ref. 66 in line 272: according to the sentence, the authors contradict themselves. 
AU: We rephrased the text for better clarity.
-       Line 278: please check, I think Wang et al. found Peptoniphilus, not Bicalho et al.
AU: It is now specified which results belong to both and which belong only to Wang et al.
-       Line 283: ref. 69 is not correct, please leave just ref. 68.
AU: It is now corrected.
-       Line 292: ref. 72 is not correct.
AU: It is removed.
-       Line 315: Treatment with PGF2alpha did not affect prevalence of SCE either at 35 or at 49 DIM (21 DIM not mentioned).
AU: The text s corrected.
-       Line 364, ref. 85: I believe this work deals with antibiotic therapy and not with alternative treatment methods.
AU: This is removed from here.
-       Line 449: blood neutrophil gene expression.
AU: It is corrected.
-       Line 484: I think the reference is wrong, it is 117 and not 118, please check.
AU: It is corrected.

Round 2

Reviewer 1 Report

Comments and Suggestions for Authors

the author has responded to all the questions the reviewer raised.

line114, “with at least 1.0 ...",1.0 or 1?

Author Response

Thank you for your review and suggestions. Regarding your question, it is rather 1. It is now corrected.

Reviewer 2 Report

Comments and Suggestions for Authors

Thank you for your work on this manuscript. There are a few places that still need to be improved.

Line 36 & 317: what is the meaning of decreased hazards of pregnancy? The context would point toward a decreased pregnancy and calving rate.

Line 42-43: having appeared - better stated as ... with an up-to-date review of research published during the last 20 years...

Line 53: Without a biopsy/cytology for differentiation, by definition metritis is differentiated clinically by the presence of systemic illness. So, is "Grade 1" better defined as endometritis?

Line 79-80: What do you mean by "dynamically alternates with clearance"? Is this statement referenced?

Line 99-100: reference?

Line 123, 128-9: These sentences are quite confusing. Of course metritis is associated with endometritis. How could the deeper structures of the uterine wall be infected/inflamed without the endometrium also being involved?

Line 207: transrectal palpation...

Line 229-230: If there is a resident vaginal and uterine microbiome, Westermann et al,. failed to detect the microbes, not that there was none present.

 Line 314: missing a comma

Line 337-338: ...found unhealthy... what do you mean?

Comments on the Quality of English Language

The manuscript needs work in this area

Author Response

Thank you for your review and your comments and suggestions.

Line 36 & 317: what is the meaning of decreased hazards of pregnancy? The context would point toward a decreased pregnancy and calving rate.

AU: Yes, you are right. The text is corrected.

Line 42-43: having appeared - better stated as ... with an up-to-date review of research published during the last 20 years...

AU: Yes, thank you. It sounds much better. The text is corrected accordingly.

Line 53: Without a biopsy/cytology for differentiation, by definition metritis is differentiated clinically by the presence of systemic illness. So, is "Grade 1" better defined as endometritis?

AU: According to the definitions by Sheldon et al. (2006; 2009), the main difference is the time after calving and the affected uterine wall parts. Yes, without a biopsy, it is hard to differentiate grade 1 metritis from endometritis in the praxis. However, we still prefer to stick to Sheldon's definitions.

Line 79-80: What do you mean by "dynamically alternates with clearance"? Is this statement referenced?

AU: Yes, it is referenced with [8]. Contamination alternating with clearance means there might be bacterial contamination, then clearance and re-contamination of the uterine lumen during the first few weeks after calving.

Line 99-100: reference?

AU: The next two sentences explain why, and all statements belong to the reference [18]:

"There is a difference between the efficacy of the defence mechanisms of primiparous versus multiparous cows. Older cows have lower uterine elasticity and slower involution than younger ones. On the other hand, in young animals, their limited exposure to bacterial contamination de lays proper immune response, while older cows may eliminate pathogenic bacteria faster [18]."

Line 123, 128-9: These sentences are quite confusing. Of course metritis is associated with endometritis. How could the deeper structures of the uterine wall be infected/inflamed without the endometrium also being involved?

AU: According to Sheldon's definitions, the main difference is the time after calving and the affected uterine wall parts, endometritis being more long-tem. In light of this, if there was a metritis after calving, some healing might happen, but there may be some endometritis remaining. We think that during the metritis, the endometrium is necessarily affected. Consequently, there is also endometritis during this period, you are right.

Line 207: transrectal palpation...

AU: It is corrected, thank you.

Line 229-230: If there is a resident vaginal and uterine microbiome, Westermann et al,. failed to detect the microbes, not that there was none present.

AU: They say: "In 45 samples (19.6%) bacteria were not found." Additionally, they checked the endometrium with cytobrush, not the vagina. We clarified the text.

 Line 314: missing a comma

AU: Thank you, it is corrected.

Line 337-338: ...found unhealthy... what do you mean?

AU: In the paper cited, the authors used the word "clearing" in the context of cervicovaginal mucus, so we corrected our wording accordingly. This interesting paper is not too detailed, however. Unfortunately, we do not know what "clearing" exactly means here. 

Reviewer 4 Report

Comments and Suggestions for Authors

Although I still think that the paper cannot make a decisive contribution to the field, as it has been explored by countless researchers and experts who have produced a huge scientific literature on the subject, I appreciated the effort to cover such a huge and difficult topic.

I still sense a ‘didactic style’ throughout the paper, but this is probably unavoidable when trying to summarize such a multi-layered topic in just a few pages.

Some highlighted parts have not been substantially changed, but overall all critical points brought to the Authors’ attention have been addressed.

Author Response

Thank you for your efforts in reading our manuscript and the benign comments. We also feel the text has a little ‘didactic style’ throughout the paper, but as you said, this is "unavoidable when trying to summarize such a multi-layered topic in just a few pages".